# Effectiveness of Whole-Body Vibration Combined with Multicomponent Training on the Risk of Falls and Quality of Life in Elderly Women with Osteoporosis: Study Protocol for a Randomized Controlled Clinical Trial

**DOI:** 10.3390/biology11020266

**Published:** 2022-02-08

**Authors:** Rúbia Rayanne Souto Braz, Shirley Lima Campos, Débora Wanderley Villela, Gabriel Barreto Antonino, Pâmella Karolline Araújo Batista, Marcelo Renato Guerino, François Talles Medeiros Rodrigues, Kennedy Freitas Pereira Alves, João Victor Torres Duarte, Diana de Andrade Silva, Daniel Florentino Lima, Arthur Felipe Freire da Silva, Karla Cybele Vieira de Oliveira, Edy Kattarine Dias dos Santos, Wagner Souza Leite, Larissa Coutinho de Lucena, Ana Paula de Lima Ferreira, Kátia Monte-Silva, Maria das Graças Rodrigues de Araújo, Redha Taiar

**Affiliations:** 1Programa de Pós-Graduação em Fisioterapia, Departamento de Fisioterapia, Universidade Federal de Pernambuco (UFPE), Recife 50670-901, PE, Brazil; rubia.braz@ufpe.br (R.R.S.B.); shirley.campos@ufpe.br (S.L.C.); debora.wanderley@ufpe.br (D.W.V.); gabriel.antonino@ufpe.br (G.B.A.); kennedyfpa@hotmail.com (K.F.P.A.); karlacvo2003@yahoo.com.br (K.C.V.d.O.); edy.santos@ufpe.br (E.K.D.d.S.); ana.lferreira@ufpe.br (A.P.d.L.F.); monte.silvakk@gmail.com (K.M.-S.); 2Departamento de Fisioterapia, Universidade Federal de Pernambuco (UFPE), Recife 50670-901, PE, Brazil; joao.tduarte@ufpe.br (J.V.T.D.); diannaandrade74@gmail.com (D.d.A.S.); daniel.florentino.lima@gmail.com (D.F.L.); arthurfelipe.ufpe@gmail.com (A.F.F.d.S.); 3Programa de Pós Graduação de Biologia Aplicada à Saúde, Universidade Federal de Pernambuco (UFPE), Recife 50670-901, PE, Brazil; wagnerszleite@gmail.com; 4Programa de Pós-Graduação em Neuropsiquiatria e Ciências do Comportamento, Universidade Federal de Pernambuco (UFPE), Recife 50670-901, PE, Brazil; 5Instituto de Medicina Integral Prof. Fernando Figueira, Recife, 50070-550, PE, Brazil; pamella_karolline@hotmail.com; 6Programa de Pós-Graduação em Saúde Translacional, Centro de Ciências Médicas, Universidade Federal de Pernambuco (UFPE), Recife 50670-901, PE, Brazil; marcelo.guerino@ufpe.br; 7Programa de Pós-Graduação em Ciências da Saúde, Universidade Federal de Pernambuco (UFPE), Recife 50670-901, PE, Brazil; 8Programa de Pós-Graduação em Ciências da Saúde, Instituto de Ciências Biológicas, Universidade de Pernambuco (UPE), Recife 50100-010, PE, Brazil; francoismedeirosfisiot@gmail.com; 9UNIESP Centro Universitário, João Pessoa 58410-407, PB, Brazil; larissacoutinho@gmail.com; 10MATIM, Université de Reims Champagne-Ardenne, 51100 Reims, France; redha.taiar@univ-reims.fr

**Keywords:** falls, vibration, osteoporosis, exercise, elderly, quality of life

## Abstract

**Simple Summary:**

Risk of falls secondary to osteoporosis consists an important problem that affects society and there is no defined treatment to ameliorate both the symptoms and quality of life of elderly women with osteoporosis and reduce the risk of falls. We aim to evaluate the effectiveness of whole-body vibration protocol combined with various exercises specifically in this population, therefore, our results might identify a defined protocol to reduce their risk of falls as strength, balance, and functional capacity improves.

**Abstract:**

Osteoporosis and the risk of falls increase the risk of fractures and events of falls. Prescriptions and programs for different forms of exercise have different impacts on the risk of falls, and exercises from multiple categories of whole-body vibration can be effective. This study aims to evaluate the effectiveness of whole-body vibration (WBV) protocol combined with multicomponent training (MCT) in elderly women with osteoporosis and their history of falls. Our proposal is a protocol for a randomized clinical trial, divided into two stages: First, development of a protocol for WVB combined with MCT for elderly women with osteoporosis and a history of falls, under the Guidelines of the American College of Sports Medicine, and following the recommendations of the Standard Protocol Items Recommendations for Interventional Trials (SPIRIT), and second, a randomized controlled clinical trial following the Consolidated Standards of Reporting Trials (CONSORT). This trial will have implications for the effectiveness of a vibration protocol combined with multicomponent exercise on the risk of falls and quality of life for older women with osteoporosis. We expect that adding full-body vibration to an exercise protocol will decrease the risk of falls and improve participants’ quality of life, as well as their strength, balance, and functional capacity.

## 1. Introduction

According to the World Health Organization (WHO), osteoporosis (OST) is defined as the loss of bone mineral density exceeding −2.5 standard deviations (T-score) concerning young adults [1]. About 200 million people worldwide have OST [2] and, with the growth of the elderly population, by the year 2050, we expect an increase of up to three times in the risk of having OST [3]. OST is considered a systemic disease whose main alterations are the decrease in bone mass and strength, with deterioration of the bone microarchitecture, increasing the chances of fractures. Women are more susceptible to OST than men, because, in addition to menopause and estrogen deficiencies, they have lower bone mineral density, leading to a greater risk of falls and imbalance problems [4]. In addition to the bone loss, other functional declines associated with aging may be present [5], such as the risk of falls, muscle strength, gait, balance, and vision that should be included in OST assessment in elderly women [6].

Women with OST have intrinsic factors that increase their risk of falls [7]. For this reason, several studies [8,9,10,11,12,13,14] investigate the effects of different therapeutic modalities in preventing the risk of falls and improving the perception of quality of life in elderly women with OST.

Studies [15,16,17] indicate that multicomponent exercises reduce the incidence and risk of falls and present few adverse events for the elderly population.

Data from a meta-analysis [18] show that long term effects of exercise reduce the risk of falls (RR 0.88; 95% CI, 0.79–0.98; 4420 participants, 20 randomized controlled trials; moderate heterogeneity) and fall injuries (RR 0.74; 95% CI, 0.62–0.88; 8410 participants, nine RCTs; moderate but not substantial heterogeneity) in the elderly. This study also points out that the most used exercises are multicomponent training (MCT), three times a week, with an average of 50 min per session. Similarly, another review [19] shows that the MCT, among them the functional ones and the balance and resistance ones, reduce the rate of falls by 34% (RaR 0.66, 95% CI 0, 50 to 0.88; 1374 participants, 11 studies; evidence of moderate certainty) and reduce the number of people experiencing one or more falls by 22% (RR 0.78, 95% CI 0.64 to 0.96; 1623 participants, 17 studies; evidence of moderate certainty).

Another therapeutic modality used in people with OST is whole-body vibration (WBV), a non-invasive and non-pharmacological intervention that can promote a reduction in the impact on bone mass and density, helping to gain balance and muscle strength and reducing the number and risk of falls [20,21,22,23,24]. WBV alone or in combination with exercise can also decrease the number of falls and the risk of falls, being essential to prevent bone fractures and osteoporosis [21,22,25].

In clinical practice, protocols with the prescription of multicomponent exercises and vibration in the elderly and patients with osteoporosis are described in isolation or combination during the use of the platform. To our knowledge, no studies have been found using WBV combined with MCT in elderly women with OST, in which the protocol contained detailed information on parameters, frequency, and duration, among others, necessary for clinical reproducibility. Given the above, the present study aims to develop a protocol for a clinical trial to verify the efficacy of WBV combined with MCT in the risk of falls and the quality of life of elderly women with OST.

## 2. Materials and Methods

### 2.1. Study Design

The following is a methodological type study to create a protocol and carry out a randomized and controlled clinical trial. It was divided into 2 stages: (1) Development of a protocol for WBV combined with MCT, following the Guidelines of the American College of Sports Medicine (ACSM) [26] for elderly women with OST with a history of falls. The protocol was developed following the recommendations of the SPIRIT statement; (2) proposal of a randomized controlled clinical trial following the norms of the Consolidated Standards of Reporting Trials (CONSORT).

#### 2.1.1. Stage 1: Protocol

At this stage, the Guidelines of the American College of Sports Medicine [26] were consulted, aiming to establish the main exercises for elderly people with OST and the risk of falls. Following these guidelines, specific intensity, frequency, exercise modality, and duration parameters were established for the profile of the participants (Table 1 and Table 2).

According to the ACSM guidelines a training program is designed to achieve individual health and fitness goals through the principles of prescribing an exercise routine. They continue to infer the importance of employing the FITT-VP principle of prescribing an exercise routine, which comprises frequency, intensity, time, type, and total volume of progression, based on ACSM evidence. In the context of individuals with clinical problems and with reservations, this prescription is subject to modification. In this context, the protocol followed the FITT Recommendations for elder individuals and with OST, following the ACSM Guidelines for Stress Tests and Prescription [26].

#### 2.1.2. Stage 2: Clinical Trial

We will develop a double-blind controlled clinical trial following the guidelines established in the CONSORT, registered in ReBec (n. RBR-3xdf4k). The following topics refer to the steps to be taken to conduct the clinical trial:

### 2.2. Participants

Participants will be recruited from the local community. Inclusion criteria are: (1) Elderly women over 60 years old with OST (diagnosed with Bone Densitometry Exam); (2) history of falls (at least two episodes in the last 12 months); (3) medical statement allowing the practice of physical exercise (cardiologic statement); (4) not participating in other research studies. Patients with: (1) Neurological diseases; (2) cognitive deficit (cognitive ability to respond and perform the exercises assessed by the Mini-Mental State Examination; MMSE), elaborated [26] and translation/modification proposed [27], using as a cutoff point for illiterate individuals = 19 points [28,29]; (3) severe Visual Impairment; (4) circulatory diseases; amputation or use of lower limb prosthesis; (5) history of recent fracture in the upper and lower limbs; (6) hernia, discopathy or severe spondylosis; and (7) neoplasm.

### 2.3. Recruitment

The project will be developed at the Laboratory of Kinesiotherapy and Manual Therapeutic Resources (Laboratório de Cinesioterapia e Recursos Terapêuticos Manuais; LACIRTEM) and at the Clinic School of the Department of Physiotherapy (Departamento de Fisioterapia; DEFISIO) at the Universidade Federal de Pernambuco (UFPE). Participants will be recruited at the Clinic School of DEFISIO, at the outpatient clinic of Clinic Hospital (Hospital das Clínicas; HC), at the Elderly Care Center (Núcleo de atenção ao idoso; NAI), at the Open University for the Elderly (*Universidade Aberta à terceira Idade*; UNATI-UFPE), and through dissemination on social and events aimed at the elderly. Those eligible and who express the intention to participate in the study, by signing the Free Consent Term, will undergo the initial interview after this recruitment.

### 2.4. Randomization and Allocation

Eligible participants will be randomized, by a researcher that is not involved in this study, through the Randomization website (www.randomization.com) into two groups: MCTV (intervention group) and MCT group (control group). Afterward, the list will be kept confidential, in which the order of the participants will be determined through numbered, sealed, and opaque envelopes that will only be opened when the intervention begins. The flowchart of participants and randomization is displayed, as shown in Figure 1.

### 2.5. Blinding

Outcome assessments will be performed by three researchers who are blind to the intervention group to which the participant was allocated. Statistical analysis will also be performed by a researcher who is blind to other research procedures.

### 2.6. Intervention

The intervention protocol will be applied by an experienced physical therapist, blind to the assessment of outcomes, soon after the enrolment period. It will consist of 8 weeks of training for both groups, totaling 24 sessions distributed three times a week, on alternating days. Each session will last between 45 and 48 min, depending on the participant’s group (Table 3).

#### 2.6.1. MCTV Group Intervention Description

This group will carry out the proposed WBV protocol combined with the MCT.

Multicomponent training should include cardiorespiratory and aerobic endurance; strength and flexibility; balance and body stability [30], flexibility not being mandatory as an MCT adequacy. Training should last 45 min, with 10 min warm-up, 30 min of the circuit, and 5 min of cool down [31,32]. It will consist of three stations: 1. cardiorespiratory and aerobic resistance; 2. strength, endurance, and flexibility; and 3. balance and body stability. The multicomponent stations will be distributed progressively as follows: (I) First station: free walking exercises will be performed on a treadmill; (II) second station: strengthening of upper and lower limbs with load progression will be performed; trunk and abdominal training and exercises in diagonal movements of the upper limbs; (III) third station: balance progression exercises will be performed, evolving different supports in orthostatic position (bilateral or unilateral), with a change of base and support surface (with and without eye-opening; in soil and foam) and balance training with a circuit.

The MCT will follow the prescribing principles under the ACSM Guidelines regarding the intensity, frequency, modality, and duration of each. The intensity during training will be regulated using the Borg Rating of Perceived Exertion scale, through which it is possible to measure the perceived exertion rate during exercise. The scale ranges from 6–20 points [33,34]. According to the individual’s fitness level, the exercises will be performed with a perceived effort between 10 and 16 points, respecting a weekly progression.

During the intervention, peripheral oxygen saturation (SpO2), heart rate (HR), and blood pressure (BP) will also be monitored.

For the aerobic exercise intensity, an incremental protocol will be used, with increased intensity every week, according to the perceived exertion rate. The intensity of the strengthening training stimulus will be incremental, with a slight to strong intensity variation as the weeks progress. It will be initially set at 40% of 1 RM (maximum repetition), with variation up to 70% of 1 RM. The intensity will be gradually progressed, as the weeks progress, being distributed as follows: one set of 10 repetitions in the first week, going to two sets of 10 to 15 repetitions

Balance activities will include static and dynamic exercises with the progression of postures as training progresses. All exercises in the program will be performed slowly to ensure the subjects’ safety, emphasizing good posture, social interaction, and pleasure [25].

Participants will perform the WBV on a vibrating platform after the end of the TMC. The vibration training will be carried out on the vibration platform KIKOS P204–110V (São Paulo, Brazil), whose vibration is of the lateral–lateral and oscillatory type. There will be 24 vibration sessions with an incremental increase of progressive frequency from 12.6 Hz to 30 Hz, the oscillation amplitude of 4 mm from peak to peak, lasting 60 s, and with a rest time of 10 to 30 s. Participants must, at the time of vibration, for better fixation of the feet on the board, perform the vibration with a silicone pad for the heels (made of non-slip material), so that only the forefoot and midfoot are in direct contact with the platform, to reduce the transmission of vibrations to the head, and a squat position of 100° will be adopted (considering 180° as total knee extension) in Figure 2. 

#### 2.6.2. MCT Group Intervention Description

The participants of the control group will only perform the MCT during the 8 weeks, totaling 24 sessions, following the same descriptions of the MCT of the GTMV group.

#### 2.6.3. Discontinuity Criteria

Participants who have three consecutive absences during the 8-week intervention period, as well as those who no longer wish to participate in the study, will be discontinued from the study. Patients who are discontinued from the study will be included in the intention-to-treat analysis.

#### 2.6.4. Treatment Adherence

Some strategies will be adopted to guarantee the adherence and attendance of these participants, such as constant communication with the patient to clarify doubts, possible questions and guidance about the OST and the risk of falls; individualized monitoring of sessions, with scheduled service, motivating the participant to complete the total number of sessions.

### 2.7. Outcome Assessments

The following variables will be collected to establish the general characteristics of the sample: weight, height, Body Mass Index (BMI), sex, marital status, place of birth, education, occupation, family income, number of chronic conditions, use of medications, and frequency of falls in the last year.

#### 2.7.1. Primary Results

Assessment of the risk of falls was performed using the Falls Efficacy Scale International (FES I), an instrument validated for the Brazilian elderly [35]. The internal consistency of the FES-I Brazil was α = 0.93, and the reliability was ICC = 0.84 and 0.91 (intra- and inter-examiners, respectively). A 16-item questionnaire will be answered on a four-point Likert scale (1 = not at all worried, 2 = a little worried, 3 = very worried 4 = extremely worried), totaling 16 items scored on a four-point scale (1 = not at all worried 4 = very worried) [36,37]. This should be reliable to assess fear of falling in older women at higher risk of falling and that the FES I score should change by >3 points or 11% to identify clinical change over time at the group level and the individual level, the sum of the score would have to change by >8 points or 29% [38].

Quality of life will be measured using the WHOQOL-OLD, validated for Portuguese. It comprises 24 items recorded on a five-point Likert scale, divided into six facets: “sensory functioning”, “autonomy”, “past, present, and future activities”, “social participation”, “death and dying”, and “intimacy”. The instrument’s overall score determines that higher scores represent better quality of life in the facets. The instrument has satisfactory characteristics of internal consistency (Cronbach’s coefficients from 0.71 to 0.88), discriminant validity (*p* < 0.01), concurrent validity (Correlation coefficients between −0.61 and −0.50), test–retest reliability (Correlation coefficients between 0.58 to 0.82), usefulness, and good psychometric performance [39,40]. The minimal clinically important differences for quality-of-life measures are variable depending on the method applied and the clinical context of the study [41].

#### 2.7.2. Secondary Results

The secondary outcomes will be measured to assess the participants’ functional capacity: balance, lower limb muscle strength, functional gait mobility, and global perception of the protocol. Treatment adherence and adverse events will also be analyzed.

The balance will be assessed using the functional range test (FRT). A reliable and valid measure of balance, which measures the ability to advance in an anticipatory postural adjustment task [42,43]. The test consists of asking the subject to stand with the right shoulder close to a wall, performing an anterior flexion of the arm at 90º, with the fingers of the hands in flexion; then, the length of the right upper limb will be registered on the ruler. After this procedure, the subject is asked to try to reach some object in front of him without taking steps or performing any compensatory strategy [44]. When developing the test, established as normative values for men 33.4 ± 3.9 cm and women 26.5± 8.9 cm [42] and in a recent meta-analysis, [44] Rosa, Perracini, and Rici (2019) determined as normative for the community elderly population values of 26.60 cm (CI 95%: 25.14–28.6 cm), while for the Brazilian population [45] established that women between 41–69 years have a mean RF of 28.54 (SD 3.61) and between 70–87 years have a mean of 27.13 (SD 2.83).

The dynamic balance of the participants will also be evaluated through the Biodex Balance System platform (Biodex Medical, USA) in the Falls Risk Test (FRt) protocol, the platform is unstable and estimates the fall risk index. The protocol is composed of 12 levels of dynamic stability with level 12 being the most rigid and level 1 being the most unstable. The participant will be instructed to stay on the platform: standing, barefoot, and with eyes open. They should maintain a vertical projection with their center of gravity in the center of the platform, observing a vertical screen located in front of their face, then the average result of three tests will be obtained, each evaluation will last 20 s, with rest periods of 10 s between them. The test will generate a General Stability Index (IGE)/Overall stability index (OSI), i.e., the total variation of the displacement from the center of the platform through the general variation of the Anterior–Posterior Stability Index (APSI) and the Medial–Lateral Stability Index (MLSI) [46,47].

Muscle strength will be assessed using the 1 Repetition Maximum (1 RM) test, which has been considered a standard for the assessment of dynamic strength [26,48,49], with adequate familiarization being a reliable indicator of muscle strength [50]. The test will be carried out in a fixed muscle-strengthening device (Mega II Movement), if necessary, also using dumbbells (to adapt the gradation of smaller loads), and the maximum amount of weight lifted will be recorded during the performance of the extension exercise, given that the strength of the extensors is a good measure to identify elderly people at high risk of falls [51].

To assess functional capacity, the distance walked in the 6-min walk test (6MWT) will be used, which adequately reflects the physical capacity of patients to perform routine tasks. The test will be carried out in an external corridor, 30 m long, with a smooth surface marked meter by meter, in which the patient must go and return as often as possible during the pre-established time. The circuit delimitation will be indicated by signaling cones. Immediately before starting the tests, participants will receive guidance, following the standardization suggested by the American Thoracic Society (ATS) [52,53,54]. For the Brazilian population, the authors investigated predicted values to determine the distance covered by a Brazilian population of adults and elderly people, which is represented by: 6MWD = 622.461−(1.846 × age) + (61.503 × male gender = 1, women = 0) and the MCID estimate for 6MWD adopted is 17.8 m when analyzing changes in interventions in moderately frail Asian elderly with fear of falling after an exercise protocol [55].

The global impression of patients will be assessed by the Patient Global Impression of Change (PGIC), adapted and validated for the Portuguese population [56]. This is an understandable, adequate, easy, and quick use instrument, being an aid in the comparison of results between interventions and/or in identifying clinically important differences. It is a one-dimensional measure in which users rate their improvement combined with the intervention on a seven-item scale ranging from “no change (or the condition got worse)” to “much better, and with a considerable improvement that made all the difference” [57].

Adherence will be calculated by the percentage of sessions completed out of the total number of prescribed intervention days (24 sessions = 100%), reported by the researcher. Adverse effects will be counted from the researcher’s record.

After the initial assessment, baseline (t0), there will be three moments for reassessment: after the 8th session (t1), 16th session (t2), and 24th session (t3).

### 2.8. Sample Size

The sample size will be based on the primary outcome of the Falls Efficacy Scale International (FES-I) as a primary outcome measure to determine the effect of the VCI approach. It will be calculated using the G*Power 3.1 program. The a posteriori sample calculation will consider a statistical power of 80% and a significance level of 5% to obtain a clinically relevant change between groups with more than 2.9 change points [38]. The calculation will also assume that 20% of participants drop out of intervention studies.

### 2.9. Statistical Method

Data will be collected and added to a database in Excel XP 2010 Microsoft ^®^ as it is collected. In parallel, the collected data will be stored and analyzed electronically using the Statistics Package for the Social Sciences (SPSS) program, version 20.00 for Windows (SPSS Inc, Chicago, IL, USA), a significance level of 95% will be adopted (*p* < 0.05) for all analyses.

Initially, a descriptive analysis of the variables will be performed using frequencies for categorical variables and mean and standard deviation for continuous variables, to characterize the sample and other analyses. Then, the verification of the normality of the data will be carried out. The behavior of quantitative variables will be performed using mean and confidence interval and if the distribution is not normal, the median and interquartile interval will be considered. Qualitative variables will be analyzed through measures of absolute and relative frequencies. Analyses will follow the intention-to-treat principle.

## 3. Discussion

To determine the efficacy of WBV combined with MCT in women with osteoporosis and a previous history of falls, an intervention study will be conducted. It is assumed that adding vibration to MCT will decrease the risk of falling and improve the quality of life of the participants.

Models of care for OST include fractures, assessments, education, and exercise programs [10]. Bone health is essential for the elderly and concerning OST, recent evidence indicates that exercise can delay its onset and reduce the risk of falling [26].

A meta-analysis of patients with OST (RR 0.832; CI 0.762–0.909; n: 3569 participants; 10 randomized clinical trials; I2 = 13.4%) observed the effects of exercise in fall-related injuries, thus demonstrating that participants at high risk of falls or with OST would benefit primarily from exercise intervention [8].

Different pieces of evidence bring the importance of exercises and WBV. However, to date, there are few studies with WBV programs added to multicomponent exercises, aimed at improving the risk of falls and the quality of life of women with OST, among these, a randomized clinical trial [58] (n:42 elderly) of a vibration protocol added to exercise on the risk of falls and quality of life observed improvements in balance, gait, and quality of life in the participants undergoing the intervention. In turn [59] in a randomized clinical trial (n:77 elderly) of WBV added to the strength and balance exercises found that the improvement in functional mobility of this WBV addition was greater than in the group that performed the exercise alone.

The MCT refers to the combination of three or more training modalities that involve exercises that involve muscular endurance, aerobic, balance, and flexibility exercises in the same training session [30,60,61]. Given the above, a multicomponent exercise program is recommended to improve physical fitness and reduce the incidence and risk of falls in the elderly [62].

According to authors [63], a meta-analysis of randomized clinical trials found that MCT improves strength and functional performance in the elderly, proving to be a good training strategy for the elderly. In ECR, [64] also verified the improvement of muscular endurance, agility, and dynamic balance in the elderly. A positive effect was demonstrated [61] of CMD and positive health outcomes for the elderly, such as cardiorespiratory fitness, functional skills (muscle strength, balance and gait skills, flexibility, and exercise capacity), in addition, to contribute to reducing the risk of falling and in some aspects of the quality of life of the elderly.

The WBV in patients with OST presents results of increased muscle strength and increased neuromuscular coordination, which can reduce the risk of falls that are often the result of fracture [65], also corroborate the positive findings of clinical trials of WBV in the elderly in improving balance [66], in improving mobility and gait performance [67] in improving different muscle strength proxies, as symmetrical maximum voluntary contraction, dynamic strength, rate of development, and isometric strength of knee extensors [68,69].

It is common knowledge that the high costs and demands of services necessary to manage patients with a history of falls and other complications arising from such a process are common. In addition to this issue, it is known that OST, when left untreated, can bring functional changes in this elderly population. This protocol aims to provide participants and the community in general with skills that provide active and healthy aging, seeking to ensure greater autonomy and independence for this population. So, it is expected that, through this protocol, benefits are possible to alleviate restrictions and promote a more active and functional life.

### Limitations

Some limitations of this protocol should be considered, which may influence the interpretation of the study’s findings, such as patient adherence to the 24-week protocol and the absence of a control group in the study—which does not perform any type of therapy, due to lack this profile of participants since the current guidelines for patients with osteoporosis include the practice of physical activities.

## 4. Conclusions

The development of that protocol based on the addition of whole-body vibration to multi-component training seeks to reduce the risk of falls and improve the quality of life of elderly women with osteoporosis and a history of falls, also hoping to ensure improvements in balance activities, strength, and functional performance, ensuring the safety and social interaction of participating patients. It is essential to infer that the results will have important implications for prevention, non-pharmacological treatment, clinical practice, and health policies, as well as that the elaboration of a detailed and targeted prescription for this target audience, allows its use in current clinical practices or future.

## Figures and Tables

**Figure 1 biology-11-00266-f001:**
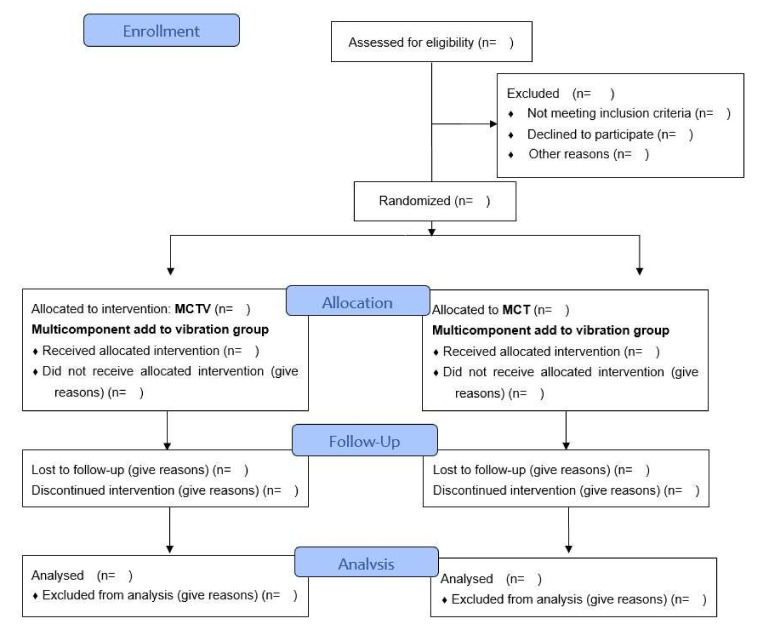
Consolidated Standards of Reporting Trials flow diagram.

**Figure 2 biology-11-00266-f002:**
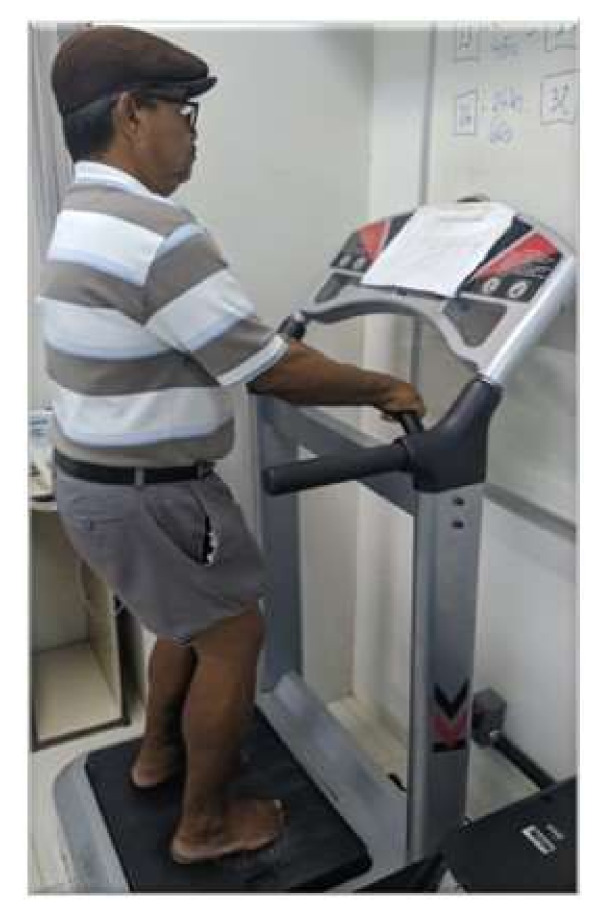
Participant position during vibration platform.

**Table 1 biology-11-00266-t001:** Multicomponent training protocol model.

Training Session	Exercise	Prescription
Aerobic/cardiorespiratory resistance	Free Walk (1st week)	10 min with an increment of 3–4 CR10 or Borg/per session
	Increase Treadmill walk (2nd–8th week)	
Resistance	Shoulder lift	**Load intensity—1RM**
	Shoulder abduction	1st and 2nd week with slight intensity (40% RM)
	Shoulder girdle dissociation	3rd and 4th week with slight intensity (50% RM)
	Abdominal exercise	5th and 6th week with moderate intensity (60% RM)
	Straight leg lift	7th and 8th week with strong intensity (70% RM)
	Hip adduction	**Frequency**
	Hip abduction	1st week: 1 set of 10 repetitions
	Knee extension	2nd and 3rd week: 2 sets of 10 repetitions
	Ankle plantar flexion and ankle dorsiflexion	4th and 5th week: 2 sets of 12 repetitions
	Sit and get up	6th to 8th week: 2 sets of 15 repetitions
Balance	Orthostasis with bilateral support	1st and 2nd week: bi support tpt/EO/stable
	Orthostasis with unilateral support	3rd and 4th week: uni support tpt/EO/stable
	Tandem posture	5th week: uni support tpt/EO/unstable
	Tandem march and tiptoe	6th week: uni support tpt/EC/stable
		7th week: uni support tpt/EC/unstable
		8th week: without tpt/EC/unstable
		Progressive postures according to the patient, in time of 10–20 s
	Stretching: sternocleidomastoid, scalene, pectoral muscle, quadriceps, and sural triceps	The intensity with a stretch to the point of muscle strain or mild discomfort. Frequency of 2 × 30 s
Fonte: Braz, 2020		

Legend: mr: maximum repetition; uni: unilateral; bi: bilateral; tpt: therapist; eo: eyes open; ec: eyes closed.

**Table 2 biology-11-00266-t002:** Protocol of the vibratory platform model.

Session	Session per Week	Time Vibration (s)/Series	Vibration Frequency (Hz)	Amplitude (mm)	Recovery (s)	Total Time of Intervention (s)
1st	3	60/4	12,6	4	10	180
2nd	3	60/4	12,6	4	10	180
3rd	3	60/4	20	4	10	180
4th	3	60/4	20	4	10	180
5th	3	60/4	26	4	20	180
6th	3	60/4	26	4	20	180
7th	3	60/4	30	4	30	180
8th	3	60/4	30	4	30	180

Legend: mm (millimeter); s (seconds); Hz (hertz).

**Table 3 biology-11-00266-t003:** Schematic diagram of timeline of enrollment, intervention, and assessments. Adapted from SPIRIT Group 2013.

	Study period
	Recruitment	Allocation	Post- allocation	Study finalization
Time point	–t1	0	(t1) 8th session	(t2) 16 th session	(t3) 24 th session	
**Enrolment**
Eligibility screen	x					
Informed consent	x					
Allocation		x				
**Interventions**
GTMV			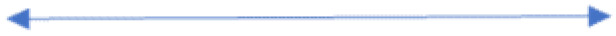	
GTM			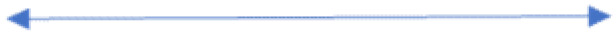	
**Assessments**
FES-I	x		x	x	x	
WHOQOL-OLD	x		x	x	x	
FRT	x		x	x	x	
FRt (BBS)	x		x	x	x	
IRM	x		x	x	x	
DTC6	x		x	x	x	
PGIC					x	
Adhrence			x	x	x	
Adverse effects			x	x	x	

Legend: after the 8th session (t1), 16th session (t2), and 24th session (t3).

## Data Availability

The datasets that will be used and/or analyzed during the current study will be made available by the corresponding author upon reasonable request.

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
