# Peer review of "Effectiveness of Whole-Body Vibration Combined with Multicomponent Training on the Risk of Falls and Quality of Life in Elderly Women with Osteoporosis: Study Protocol for a Randomized Controlled Clinical Trial"

_biology, 2022, doi:10.3390/biology11020266_

Round 1

Reviewer 1 Report

Thank you for submitting the manuscript “Effectiveness of Whole Body Vibration Associ-Ated with Mul-ticomponent Training on the Risk of Falls and Quality of Life in Elderly Women with Osteoporosis: Study Protocol for a Randomized Controlled Clinical Trial” to Biology. The work is well written, the experiment was very well designed and conducted, and the results are encouraging. Therefore, my suggestion is to accept the article with minor revisions.
1) Keywords: I suggest adding or replacing words that are not in the title.
2) Introduction: é importante deixar claro ao leitor o estado da arte dos protocolos anteriores que possuem o mesmo objetivo deste protocolo proposto. Sem uma justificativa da razão de propor um novo protocolo é impossível avaliar no que o protocolo proposto vai ser melhor que um protocolo já validado. Os autores precisam adicionar também qual o efeito metabólico dos protocolos relatados na introdução. Demonstrar somente os resultados do protocolo não me parece interessante ao leitor.
3) It would be interesting to add heart rate control to the study protocol since each individual used in the research has a time-intensity resistance in physical activity.
4) To test the efficiency of a protocol, it is normal to add an already validated methodology. However, this manuscript does not describe how validation of the proposed protocol will be performed.
Point-by-point:
Line#2: Correct “Associ-Ated”
Lines#35-41: I believe it is not part of this manuscript.
Line#45: change “is” by “was”
Line#46: change “a history” by “their history” and “this is” by “Our proposal”
Lines#51 and #52: avoid abbreviations in the abstract or include their definition
Lines#47-50: for clarity it would be better if the two stages of the search were in the same sentence. Please rewrite.
Line#62: include point after (1)
Lines#78, 82, 83, etc: include space between words
Line#93: cut “to date” and include after “however” “to our knowledge”

Author Response

Dear Reviewer,

We appreciate the kind consideration of this paper by the reviewer and the opportunity to highlight the value of our research. The paper has been improved as detailed below, based on reviewer’s thoughtful comments. We have noted our changes in “red” so they are easy to locate. We thank you sincerely for your time and consideration.

Reviewer 2 Report

Thanks for the opportunity to review this manuscript. This Study Protocol aims to examine the effectiveness of whole body vibration and multicomponent training on the risk of falls and quality of life in elderly women with osteoporosis. The proposed study is interesting. There are strengths and weaknesses. My following comments include a summary of weaknesses and offer suggestions for the authors' consideration.

  1. In Introduction, the transition from the first paragraph to the second paragraph needs to be further improved, for example, how OST would lead to balance issues and falling risk.
  2. In the fourth paragraph of the Introduction, the authors provided results from a Meta-analysis. I wonder if those studies were conducted in community-dwelling older adults and how many were women, as these are relevant to your study population.
  3. There is only one paragraph that talks about the effects of WBV and the results are very limited. Please provide more evidence of its effectiveness to better justify the importance of your proposed study.
  4. The purpose of this study is to examine the effects of the combination of WBV and MCT. It would be more appropriate to use “WBV combined with MCT” instead of “associated with”.
  5. In participant recruitment, there is a typo “ractice of physical exercise”, which I assume should be “practice…” In (4) Not participating in other surveys, please clarify if it means not participating in other research studies.
  6. In Page 4, please clarify who will perform the randomization.
  7. In Figure 1, it would be better to provide the sample size as well as the pre- and post-assessments.
  8. It seems that this study will include many outcome assessments. I wonder if one examiner is enough.
  9. On page 6 Line 196-205, that describes the WBV: “Participants must, at the time of vibration, for better fixation 201 of the feet on the board, perform the vibration with a silicone pad for the heels (made of 202 non-slip material), so that only the forefoot and midfoot are in direct contact with the 203 platform…”, it would be better to include a figure for better illustration.  
  10. In 2.6.4. The authors mention the adherence. I wonder if you will measure the participants’ adherence and fidelity to the intervention.
  11. In this study, the primary outcomes are the falls efficacy and quality of life. However, these are subjective measures. Indeed, the intervention itself focuses on resistance and balance training as shown in Table 01. Objective measures of strength and balance seem to be the direct outcomes that will reflect the training components. The improvement of falls efficacy and quality of life may be through the pathway of the improved balance and strength. From this point of view, I wonder if the primary and secondary outcomes need to be switched.
  12. In this study protocol, is there a follow-up period to observe the “wash-out” or long-term effect of the intervention?
  13. Although this study includes an outcome of falls efficacy, I wonder why number of falls or fall rate are not included as the outcome variables.
  14. The authors have included the sample size calculation, but the estimated sample size was not provided. Based on the power calculation, the desired sample size should have been achieved.
  15. In the Discussion, the authors mentioned that they expect to see the decreased risk of falling and quality of life, but again these are subjective measures. Please consider my suggestion in comment #11 or provide a strong justification.
  16. In the Limitations, are there any concerns on the safety of performing WBV and any adverse effects from the intervention?

Author Response

(The authors gave the same response as above.)

Round 2

Reviewer 2 Report

Thanks for the revised manuscript. The clarity has been improved. I have a few additional comments below: 

  1. The sentence “…being more susceptible to OST than men because in addition to going through menopause they also have a lower bone mineral density, which leads to a higher risk of falls and imbalance problems” has grammar issue. Please correct.
  2. Although reference 4 is provided, please elaborate why low bone density leads to falling risk.
  3. Line 74, “…other functional declines combined with aging may be present”. “Combined with” is not appropriate here. Please use “associated with”.
  4. Line 109-111, “however, to our best knowledge, there is still no studies on the vibration immediately after a protocol of multicomponent exercises. However, to our knowledge no studies have been found to date using WBV combined associated with MCT in elderly women with OST” seem repeated or the expression is unclear. Please clarify.
  5. Line 206-207, the sentence is not English. Please correct it.
  6. The current SPSS version is 27. I wonder why the authors still use version 20.
  7. There are still grammar issues and mistakes in some sentences. I would suggest the authors seek help from a native English speaker to improve the clarity of the paper.

Author Response

Dear Reviewer,

We appreciate the kind consideration of this paper and the opportunity to highlight the value of our research. The paper has been improved as detailed below, based on reviewer’s thoughtful comments. We have noted our changes in “red” so they are easy to locate. We thank you sincerely for your time and consideration.

Thank you for revising the manuscript. The issues are corrected as follows:

  1. The sentence “…being more susceptible to OST than men because in addition to going through menopause they also have a lower bone mineral density, which leads to a higher risk of falls and imbalance problems” has grammar issue. Please correct.

Thank you for this observation. It is now corrected in lines 60-62.

  1. Although reference 4 is provided, please elaborate why low bone density leads to falling risk.

Thank you, it is highlighted in lines 62-66

  1. Line 74, “…other functional declines combined with aging may be present”. “Combined with” is not appropriate here. Please use “associated with”.

Thank you, it is corrected in line 63

  1. Line 109-111, “however, to our best knowledge, there is still no studies on the vibration immediately after a protocol of multicomponent exercises. However, to our knowledge no studies have been found to date using WBV combined associated with MCT in elderly women with OST” seem repeated or the expression is unclear. Please clarify.

Thank you for this thoughtful observation, it is now corrected in lines 90-92

  1. Line 206-207, the sentence is not English. Please correct it.

Thank you for this reminder, it is corrected in lines 183-184.

  1. The current SPSS version is 27. I wonder why the authors still use version 20.

Thank you for this question. The authors use the 20th version because we have the rights to use (purchase) this version.

  1. There are still grammar issues and mistakes in some sentences. I would suggest the authors seek help from a native English speaker to improve the clarity of the paper.

Thank you for this consideration. We have revised the manuscript together with an English native.
